# Could Phylogenetic Analysis Be Used for Feline Leukemia Virus (FeLV) Classification?

**DOI:** 10.3390/v14020249

**Published:** 2022-01-26

**Authors:** Lucía Cano-Ortiz, Caroline Tochetto, Paulo Michel Roehe, Ana Cláudia Franco, Dennis Maletich Junqueira

**Affiliations:** 1Virology Laboratory, Institute of Basic Health Sciences, Federal University of Rio Grande do Sul, Rua Sarmento Leite 500, Porto Alegre 90150-070, Brazil; lcanoo@unal.edu.co (L.C.-O.); caroline.ctto@gmail.com (C.T.); proehe@gmail.com (P.M.R.); 2Clinic for Gastroenterology, Hepatology, and Infectiology, Medical Faculty, Heinrich-Heine-University Düsseldorf, 40225 Düsseldorf, Germany; 3EMBRAPA Swine and Poultry, Concórdia 89715-899, Brazil; 4Centro Universitário Ritter dos Reis-UniRitter, Rua Orfanotrófio, 555, Alto Teresópolis, Porto Alegre 90840-440, Brazil

**Keywords:** FeLV, enFeLV, phylogenetics, recombination

## Abstract

The surface envelope (SU) protein determines the cell tropism and consequently the pathogenesis of the feline leukemia virus (FeLV) in felids. Recombination of exogenous FeLV (exFeLV) with endogenous retroviruses (enFeLV) allows the emergence of more pathogenic variants. Currently, phenotypic testing through interference assays is the only method to distinguish among subgroups—namely, FeLV-A, -B, -C, -E, and -T. This study proposes a new method for FeLV classification based on molecular analysis of the *SU* gene. A total of 404 publicly available SU sequences were used to reconstruct a maximum likelihood tree. However, only 63 of these sequences had available information about phenotypic tests or subgroup assignments. Two major clusters were observed: (a) clade FeLV-A, which includes FeLV-A, FeLV-C, FeLV-E, and FeLV-T sequences, and (b) clade enFeLV, which includes FeLV-B and enFeLV strains. We found that FeLV-B, FeLV-C, FeLV-E, and FeLV-T SU sequences share similarities to FeLV-A viruses and most likely arose independently through mutation or recombination from this strain. FeLV-B and FeLV-C arose from recombination between FeLV-A and enFeLV viruses, whereas FeLV-T is a monophyletic subgroup that has probably originated from FeLV-A through combined events of deletions and insertions. Unfortunately, this study could not identify polymorphisms that are specifically linked to the FeLV-E subgroup. We propose that phylogenetic and recombination analysis together can explain the current phenotypic classification of FeLV viruses.

## 1. Introduction

Feline leukemia virus (FeLV) is an exogenous gammaretrovirus horizontally transmitted among domestic and wild felids and is known to cause a range of diseases, including lymphadenopathy, anemia, bone marrow suppression, immune suppression, lymphoma, leukemia, and ultimately death [1,2,3,4]. FeLV is shed in saliva, nasal secretions, urine, feces, and milk of infected cats and is mainly transmitted horizontally through saliva, blood, and other body fluids by close contact between cats. Transmission can also take place from an infected mother cat to her kittens, either before birth or while nursing [5]. FeLV has the highest case fatality rate in domestic cats out of all major feline viruses in spite of remaining one of the few retroviral diseases with an available effective vaccine [6,7]. A clear reduction in incidence was observed in countries where the vaccine is efficiently established. Nonetheless, FeLV remains one of the most common viral agents in countries where vaccination is not a common practice [8].

enFeLV is the endogenous counterpart of exogenous FeLV and represents generally nonfunctional elements, naturally present in the genome of the genus *Felis* with different numbers of copies [4,9,10]. These endogenous elements invaded the feline genome prior to the speciation of cats and are transmitted from parent to offspring as integral components of chromosomes [11]. Despite not forming infectious particles or inducing disease in the host, the expression of enFeLV transcripts has been observed in many tissue types of healthy or FeLV-positive cats [12,13]. In addition, enFeLV seems to act on the biology of exogenous retroviral infection either increasing or decreasing susceptibility to FeLV infection [14,15,16]. FeLV and enFeLV are approximately 86% similar at the nucleotide level and can readily recombine to generate new FeLV variants [17]. 

The entry of FeLV into the host cell is mediated by the binding of the envelope glycoprotein (SU or gp70) to host receptors, which vary depending on the FeLV subgroup. Currently, FeLV classification is primarily based on phenotypic tests to evaluate receptor usage, such as viral interference assays (IAs). Six different groups have been described so far: FeLV-A, FeLV-B, FeLV-C, FeLV-D, FeLV-E, and FeLV-T [17,18]. Variations in SU protein affect the affinity to the cell receptor and can ultimately influence the clinical manifestation [19]. FeLV-A is the most commonly described subgroup of FeLV and has been reported to be less pathogenic than other FeLV subgroups [6]. FeLV-B arises through recombination events between the envelope (*env*) gene of FeLV-A and enFeLV transcripts [20]. This process occurs during retroviral transcriptase-directed DNA synthesis following copackaging of endogenous and exogenous viral genomes. Furthermore, this might allow for a change in cellular tropism and a shift in disease progression [21]. The emergence of FeLV-B following FeLV-A infection is considered to result in both higher morbidity and mortality. While FeLV-A is commonly transmitted horizontally, similar processes have only rarely been observed with FeLV-B [4,22,23]. FeLV-C is a less common subgroup that arises from de novo mutations in the *env* gene of FeLV-A and has been associated with the development of aplastic anemia [6,24]. FeLV-D is the result of recombination between exogenous viruses and domestic cat endogenous retrovirus (ERV-DC) that are divergent from enFeLV [25]. FeLV-E is a recently described subgroup that was isolated from natural thymic lymphoma in cats [26]. Ultimately, FeLV-T is a T-cytopathic virus, isolated from a cat with FeLV feline acquired immune deficiency syndrome (FeLV-FAIDS) that has emerged via the mutation of FeLV-A strains [26]. Although these recombination events and mutation in env typically have led to increased pathogenicity, the FeLV variants are associated with decreased ability for horizontal transmission requiring FeLV-A as a helper virus [1,20,25].

Viral interference assays have determined the virus biological activity and coinfection profiles of FeLV viruses [6]. Even with the advancement of sequencing technologies, these classical techniques continue to be the main method to identify historic and novel subgroups [17]. Data linking the current phenotypic classification of FeLV samples n to their phylogenetic history are lacking. Only a few studies have aimed to characterize FeLV subgroups using a phylogenetic approach [3,6,17]. The objective of this study was to investigate the phylogenetic relationship underlying the FeLV subgroups and explore more deeply the recombination process driving FeLV evolution.

## 2. Materials and Methods

### 2.1. Dataset

Partial and complete FeLV SU sequences were selected from the GenBank (http://www.ncbi.nlm.nih.gov/nucleotide/ accessed date on 21 March 2021). To ensure the selection of high-quality data, sequences were chosen only when meeting the following criteria: (a) one sequence per host, (b) minimum length of 1050 base pairs (bp), and (c) absence of premature stop codons. Nucleotide alignments were generated using the MUSCLE software [27] and visualized in AliView [28].

### 2.2. Recombination Analysis

Recombinant sequences were detected using seven different methods (RDP, GENECONV, Chimaera, MaxChi, Bootscan, SiScan, and 3Seq) in RDP4 v4.85 [29]. Default parameters were used and recombination events were only accepted if three or more of these methods detected breakpoints with significant *p*-values (*p* < 0.05). Sequences were also imported into SplitsTree4 [30] and tested for recombination using the PHI test [31]. PHI test calculates the pairwise homoplasy index (PHI) as the mean of the refined incompatibility scores obtained for nearby nucleotide sites across the sequences.

Recombinant sequences were submitted to bootscanning analysis in Simplot 3.5.1 [32] to detect putative recombination among subgroups. The Kimura 2-parameter model was used as a distance model on a sliding window of 200 nucleotides (nt) by increments of 20 nt. Recombinant samples identified by RDP4 were used as query sequences in Simplot. For the bootscan, non-recombinant sequences were used as references.

### 2.3. Phylogenetic Reconstruction

A maximum likelihood (ML) tree incorporating the best-fitted nucleotide substitution model (GTR + F + R5) was reconstructed in the IQtree web server [33]. The robustness of the resulting tree was evaluated by rapid bootstrapping (UFB) with 1000 pseudoreplicate datasets. The phylogenetic tree was visualized in FigTree v1.4.4 [34] and drawn using R [35].

## 3. Results

### 3.1. Dataset Curation

Four hundred and fifteen FeLV SU sequences met our criteria for data quality and were downloaded from the GenBank (1368 base pairs, Appendix A). However, only 63 of these sequences had information available on phenotypic tests or subgroup assignments (39 FeLV-A, 5 FeLV-B, 1 FeLV-C, 7 FeLV-D, 1 FeLV-E, 2 FeLV-T, and 4 enFeLV). When aligning FeLV-D samples with other FeLV/enFeLV sequences, vast differences between sequences were observed. Therefore, the phylogenetic analysis of this subgroup could not be rendered possible, resulting ultimately in the exclusion of the seven FeLV-D sequences from the following analyses. Even though the exclusion does not preclude phylogenetic analyses, the reduced number of sequences may interfere with the extent of conclusions regarding FeLV evolution.

### 3.2. Different Patterns of Recombination Were Identified

The results for the PHI test support that FeLV evolution might be modeled by recombination (*p* > 0.01). To detect putative recombinant sequences and minimize the potentially disruptive impact of recombination on phylogenetic interpretation [29,36], the dataset was analyzed in RDP4. Overall, 55 FeLV SU sequences displayed at least 1 of 24 different recombination patterns and were subsequently treated as recombinant sequences. These sequences comprise one FeLV-A, four FeLV-B, and a single FeLV-C sequence (Figure 1 and Figure 2, and Appendix A). After excluding those 55 recombinant sequences from the alignment, no significant recombination signal was detected using the PHI test (*p* = 0.8244), therefore suggesting that all recombinant sequences were effectively identified. Simplot analysis allowed for the inference of the most likely origin of each recombinant fragment (Figure 2).

### 3.3. FeLV SU Gene Can Be Divided into Two Large Clades

Phylogenetic reconstruction was performed using all FeLV samples included in this study (Figure 3, Appendix A). The incorporation of recombinant sequences in this phylogenetic analysis aims to identify the impact of recombination on FeLV subgroups. In addition, it helps clarify the evolutionary relationship between these samples. The phylogenetic tree displays the subdivision of FeLV SU sequences into two well-supported (UFB > 90) main clades (Figure 3): (a) clade FeLV-A, which includes all FeLV-A sequences intermingled with FeLV-C, FeLV-E, and FeLV-T; (b) clade enFeLV, which encompasses all FeLV-B and enFeLV sequences included in this study. The split of the dataset into two different clades is corroborated by the analysis of the *SU* gene structure (Figure 2 and Figure 3).

### 3.4. The SU Envelope Gene Has Five Variable Regions; the Pattern of the Variable Regions Is Conserved Intraclade

While most of the sequences inside clade FeLV-A present insertions in VRA and VRC regions, sequences in clade enFeLV have a 30 bp and 15 bp deletion in this region, respectively.

A similar pattern was observed in the VRB region: The 69 bp deletion is exclusively shared by viruses of the FeLV-A clade but not by viruses in the enFeLV clade. Additional variable regions (here labeled as VRD and VRE) were identified inside the proline-rich region (PRR) and in the C-domain region. VRD was exclusively observed in sequences grouping within the enFeLV clade, while VRE insertion was exhibited by 13 sequences within clade FeLV-A (Figure 3 and Appendix A).

### 3.5. The FeLV-A Clade

In total, 315 sequences clustered in clade FeLV-A, including 39 FeLV-A sequences (39/39 publicly available), 2 FeLV-T (2/2 publicly available), 1 FeLV-C (1/1 publicly available), 1 FeLV-E (1/1 publicly available), and 272 sequences without information about subgroup assignment. Seven recombinant sequences with unique patterns of recombination could be identified within this clade (Figure 1 and Appendix A): one FeLV-A (AB635503, recombination pattern 20), one FeLV-C (M14331, recombination pattern 18), and other five sequences without subgroup assignment (recombination patterns 8, 8 + 14, 10, 18, and 24). Four of these samples presented patterns of recombination that seem to exclusively include fragments derived from the FeLV-A clade, suggesting that sequences joining clade FeLV-A can recombine with themselves. Conversely, samples with recombination patterns 8 (AB635618), 8 + 14 (AB635637), and 10 (AY662460) have the C-domain derived from endogenous viruses (clade enFeLV), despite presenting most of their SU derived from clade FeLV-A.

### 3.6. The enFeLV Clade

Clade enFeLV contains 89 FeLV SU sequences, including 4 enFeLV (4/4 publicly available), 5 FeLV-B (5/5 publicly available), and 80 sequences without information on subgroup assignment (Figure 3). Among these samples, 4 FeLV-B and other 44 samples without subgroup assignment presented evidence for recombination, altogether displaying 19 different patterns of recombination. Despite being recombinants, the majority of sequences (96%) preserved the RBD region derived from endogenous strains (absence of VRA and VRC but presence of VRB). In addition, most of these samples (78%) exhibited an enFeLV profile within the PRR region (Appendix A). Together, these results suggest that recombination appears to be an important factor for the evolution of FeLV-B and enFeLV samples. This is supported by the fact that 54% of the samples within this clade are recombinant sequences. Moreover, specific signatures inside RBD and PRR regions might be critical for these viruses, with almost 80% of them presenting conserved motifs at the gene level.

### 3.7. Geographical Distribution of FeLV-A Clade

A second tree was reconstructed using only non-recombinant FeLV sequences (Appendix A). In addition to the enFeLV basal clade, this phylogeny suggests that clade FeLV-A is divided into two well-defined supported groups. Curiously, these clades group sequences from different geographical origins: While the superior clade includes only sequences from Japan, the second clade includes sequences from Brazil, the United States, and the United Kingdom. On the other hand, the non-recombinant clade enFeLV mostly grouped sequences from Japan and China (Figure 4, Appendix A). Unfortunately, FeLV sequences are not systematically collected on a global scale, therefore preventing further analysis on the relationship between phylogenetic classification and geographical distribution.

## 4. Discussion

In the last 40 years, viral interference assays have been the only method used to distinguish and define FeLV subgroups [17,36]. Analysis of the superinfection interference properties of FeLV variants has led to the identification of four interference types among FeLV subgroups. Eventually, these subtypes were associated with specific clinical phenotypes: FeLV-A uses the feline thiamine transport protein 1 (feThTr1) [37] and has been linked to macrocytic anemia, immunosuppression, and lymphoma [2]; FeLV-B utilizes the inorganic phosphate-sodium symporter fePit1 (SLC20A1), and the closely related fePit2 (SLC20A2) [38,39,40] and is tumorigenic [2]; FeLV-C uses a heme export protein called FLVCR [41,42] and has been associated with the development of aplastic anemia [43]; finally, FeLV-T requires Pit1 and a soluble cofactor expressed by endogenous FeLV-related sequences (FeLIX) [18,44] and has been connected to fatal immunodeficiency [45]. FeLV-E was only recently described and its receptor remains unidentified [26]. In spite of being the only described method to date, phenotypic tests are laborious and time consuming. This, in turn, creates a technical barrier and restricts extensive FeLV classification, ultimately interfering with molecular epidemiology or evolutionary studies. With the advancement of next-generation sequencing, genetic tests have become cheaper, faster, more reliable, and, finally, also more precise than phenotypic characterization. However, despite the clear advantages of sequencing technologies, only a few studies thus far have tried to correlate phenotypic classification to phylogenetic reconstruction for FeLV viruses [3,17,46,47]. Here, we used molecular data to characterize FeLV subgroups and outline a proposition for a new classification system based on phylogenetic analysis.

Despite being classified into different subgroups, FeLV-A, FeLV-C, FeLV-E, and FeLV-T SU sequences grouped within the same clade in the phylogenetic tree (clade FeLV-A, Figure 3). All sequences within this clade presented a similar gene organization, sharing an insertion of 30 bp in VRA and 15 bp in VRC, in addition to a 69 bp deletion in VRB and a 27 bp deletion in VRD (Figure 3). FeLV-A sequences are the most abundant strains within this cluster and were found scattered through the whole clade. Therefore, it is reasonable to assume that the most recent common ancestor of clade FeLV-A presented a gene organization related to FeLV-A and used feThTr1 as its host receptor. The inclusion of other FeLV subgroups with different host receptors, but similar gene organization within this clade may suggest that FeLV-C, FeLV-E, and FeLV-T arose in separate monophyletic events from FeLV-A viruses. This process most likely includes mutation and/or recombination. In addition, it seems that the evolution of sequences inside clade FeLV-A is somehow related to its respective geographical origin. However, insufficient data regarding sampling locations of the majority of the sequences analyzed here prevent further conclusions.

Apparently, FeLV-C has arisen by means of recombination (Figure 1). Our analysis suggests that FeLV-C is the result of a recombination event that included mostly the FeLV-A *SU* gene and only a small fragment of endogenous strains (Figure 1). The recombination breakpoint in the C domain might be the explanation for the fact that FeLV-C has a different receptor (FLVCR) than the one used by most samples grouped within the clade FeLV-A (feThTr1) [37,41]. A previous study investigating the role of the C-domain region of FeLV-C strain proposes that this region is critical for efficient SU binding. The region may function as a second receptor-binding domain, which, in addition to RBD, interacts with the host receptor to initiate virus infection [48]. The phenotypic results for FeLV-C seem to corroborate the molecular analyses performed in this study, suggesting that recombination may be the main evolutionary event defining the receptor change in this variant.

FeLV-T, in contrast, may have arisen from the accumulation of mutations in FeLV-A virions, allowing the change of host receptor from FETHTR1 to Pit1. Both FeLV-T samples included in this study were grouped in a monophyletic clade and, unlike all sequences inside clade FeLV-A, exhibited an 18 bp deletion in VRA and an 18 bp insertion in the VRE region. Even though the deletion of 18 bp in VRA is smaller, it still lies in the same position as the 30 bp deletion presented by FeLV-B viruses, which could explain the use of the same receptor (Pit1) for these two strains. Intriguingly, Shojima et al. (2006) in an ex vivo model using different cell lines proposed that the host ranges of FeLV-T and FeLV-B were not identical. Furthermore, they suggested a different Pit1 usage at the post-binding level for the two strains [49]. Another 11 sequences with no information on subgroups also presented the 18 bp deletion in VRE. Only one of these sequences (M87886) has the 18 bp insertion in the VRA and is grouped in the same monophyletic clade as FeLV-T. This unclassified sequence most likely represents another strain of FeLV-T. Together, these results support the idea that polymorphisms in VRA are notoriously more important to defining the host receptor for strains grouped within clade FeLV-A than polymorphisms in VRE.

Unfortunately, this study was not able to find genetic polymorphisms or evolutionary mechanisms in the FeLV-E strain that could explain its receptor usage. We used entropy analysis to compare FeLV-E with all FeLV-A sequences, concluding that no amino acid site could be specifically associated with the FeLV-A strain (*p* > 0.05). In addition, no recombination breakpoint was detected in this sequence. The fact that only one sequence of this subgroup has been sampled so far may have prevented any conclusions about the relationship between phenotypic testing and phylogenetic clustering.

Clade enFeLV encompassed FeLV-B, enFeLV, and most of the recombinant sequences identified in this study (87%). The similarity of the gene structure seems to be the main factor driving the clustering of these sequences. In general, these samples presented a 30 bp deletion in the VRA (100% of the sequences in the clade), a 15 bp deletion in VRC (95%), and 2 insertions of 13 and 14 bp in the VRD (88%). Despite sharing a similar structure, enFeLV and FeLV-B sequences were found segregated within the enFeLV clade in our phylogenetic analysis. Notably, FeLV-B and all the recombinant sequences without subgroup assignment were basal to a monophyletic clade that includes, among other sequences, all four enFeLV samples.

Corroborating previous studies, our analysis shows that FeLV-B arose several times through recombination between FeLV-A and endogenous strains most likely after copackaging of expressed transcripts into a single virion [50,51]. These results suggest that FeLV-B strains must include the RBD and PRR regions derived from endogenous viruses alongside a short and variable length fragment derived from exogenous strains at the 3′ end of the *SU* gene (Figure 1 and Figure 2). Pandey et al. (1991), examining exogenous/endogenous recombination during in vitro infections, have revealed that replication efficiency and cellular tropism depend on the length and region of the enFeLV sequence incorporated into the FeLV-B recombinant [52]. In addition, this study shows that amino acid changes inside VRA and VRB are the main factors for the ability of FeLV-B to bind receptor Pit1 and/or Pit2. Sample JF957363 was identified as the only non-recombinant FeLV-B sample in our analysis. However, when subjecting this sample to more in-depth analysis, we found that this isolate carries the enFeLV SU, whereas long terminal repeats (LTRs) and Gag were derived from exogenous viruses. This accounts for the relationship with the enFeLV sequences in the phylogenetic tree [53].

Frequent recombination generated FeLV strains combining endogenous and exogenous viruses. Only seven recombinant sequences were grouped within clade FeLV-A of which all maintained the RBD and PRR regions from FeLV-A strains. The C domain is, however, variable and can apparently be either derived from exogenous or endogenous viruses (Figure 1 and Appendix A). Once the RBD is completely exogenous, these recombinants probably display the same interference pattern as the exogenous FeLV-A viruses. On the other hand, clade enFeLV clustered a total of 48 recombinant sequences. Out of these sequences, the majority present the RBD and the PRR regions derived from endogenous strains. Following the pattern observed for recombinant sequences within clade FeLV-A, the C-domain region in these sequences was found to be derived from exogenous or endogenous viruses. It is interesting to highlight that the farther the sequence is from the root of the enFeLV clade in the phylogenetic tree, the greater is the size of the fragment derived from exogenous viruses in these recombinant sequences (Figure 2).

Here, we used both genetic and evolutionary approaches to explain the current phenotypic classification of FeLV viruses. We found that FeLV-B, FeLV-C, FeLV-E, and FeLV-T *SU* gene sequences share similarities to FeLV-A viruses. They most likely arose independently through mutation or recombination from the FeLV-A strain. FeLV-B is a classical product of recombination between FeLV-A and enFeLV strains that presents the RBD and PRR regions derived from endogenous viruses. Despite also arising from recombination events, FeLV-C incorporated both RBD and PRR regions from FeLV-A strains. FeLV-T is a monophyletic subgroup that probably originated from FeLV-A through combined events of deletions in the RBD region and insertions in the C domain of the *SU* gene. Unfortunately, this study could not identify polymorphisms that are specifically linked to the FeLV-E subgroup and could explain its phenotypic results for receptor usage. To conclude, we propose that the use of phylogenetic analysis can be efficiently used to classify FeLV subgroups.

## Figures and Tables

**Figure 1 viruses-14-00249-f001:**
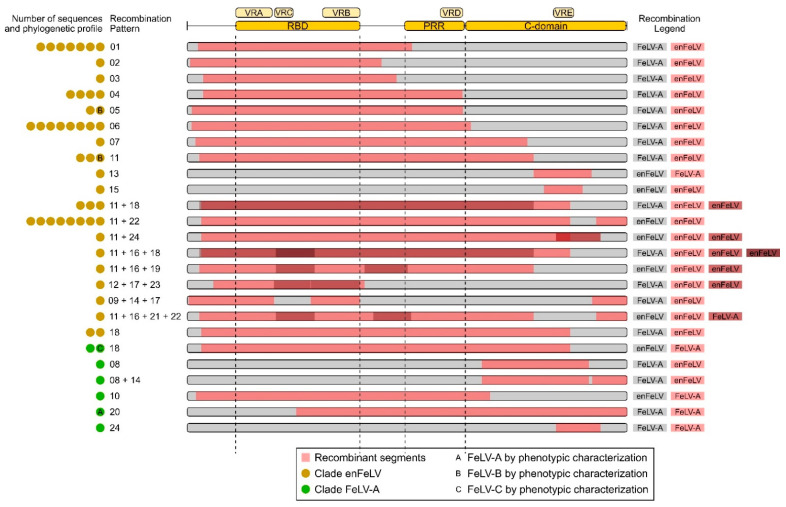
Scheme of the recombinant patterns obtained in RPD4. The number of sequences representing each recombinant pattern is indicated by colored circles. Sequences belonging to clade enFeLV are indicated in yellow and sequences belonging to clade FeLV-A in green. The recombination segments are shown for each pattern. Colored circles with letters A, B or C represent FeLV-A, FeLV-B or FeLV-C sequences identified by phenotypic characterization. RBD: receptor binding site, PRR: proline-rich region, VRA, VRC, VRB, VRD, VRE: variable region A, C, B, D and E respectively.

**Figure 2 viruses-14-00249-f002:**
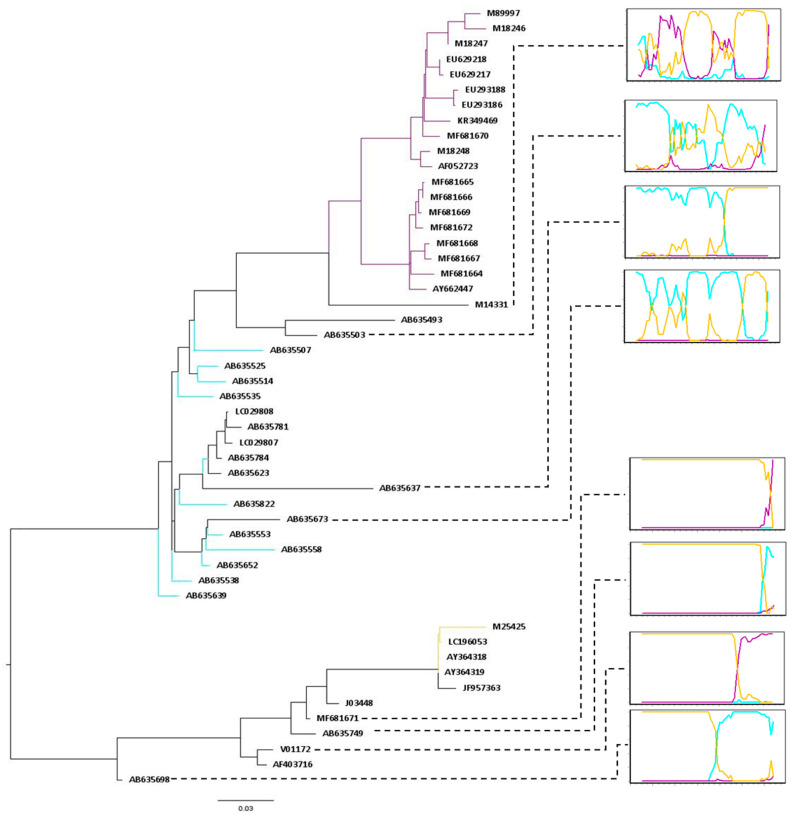
Bootscan of some recombinant sequences. Recombinant sequences were submitted to bootscanning analysis in Simplot 3.5.1 [32]. The Kimura 2-parameter model was used as a distance model on a sliding window of 200 nucleotides (nt) by increments of 20 nt. For the bootscan analysis, recombinant samples identified by RDP4 were used as query sequences in Simplot and, non-recombinant sequences were used as references. A phylogeny was reconstructed by using the best-fitted nucleotide substitution model (GTR + F + R5) in the IQtree web server with some recombinant sequences (AB635493, AB635503, AB635637, AB635638, AB635673, AB635698, AB635749, M14331, MF681671, and V01172) and non-recombinant sequences according to the RDP4 analyses. Branches of some non-recombinant sequences are colored according to the clades found in the Figure 4.

**Figure 3 viruses-14-00249-f003:**
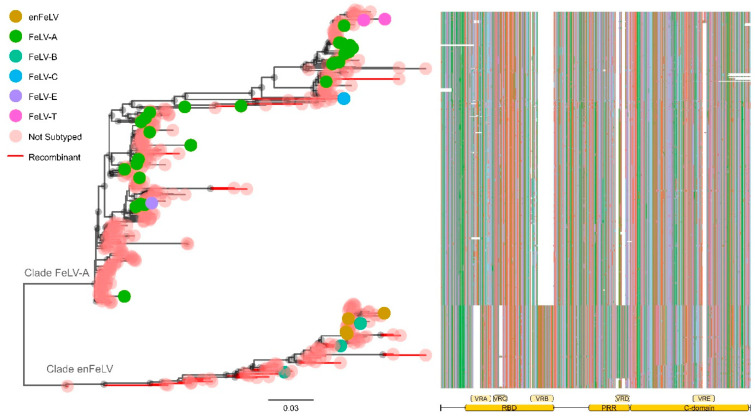
Maximum likelihood tree of 404 FeLV SU *env* sequences and alignment. Branches colored in red represent recombinant sequences detected by RDP4 software. Colored circles at the tip of the branches represent sequences with a previously defined subgroup based on interference assays. Red circles denote sequences without information on subgroup assignment: yellow: enFeLV; green: FeLV-A; blue: FeLV-A; aquamarine: FeLV-B; blue: FeLV-C; purple FeLV-E; pink: FeLV-T. RBD: receptor binding site, PRR: proline-rich region, VRA, VRC, VRB, VRD, VRE: variable region A, C, B, D and E respectively.

**Figure 4 viruses-14-00249-f004:**
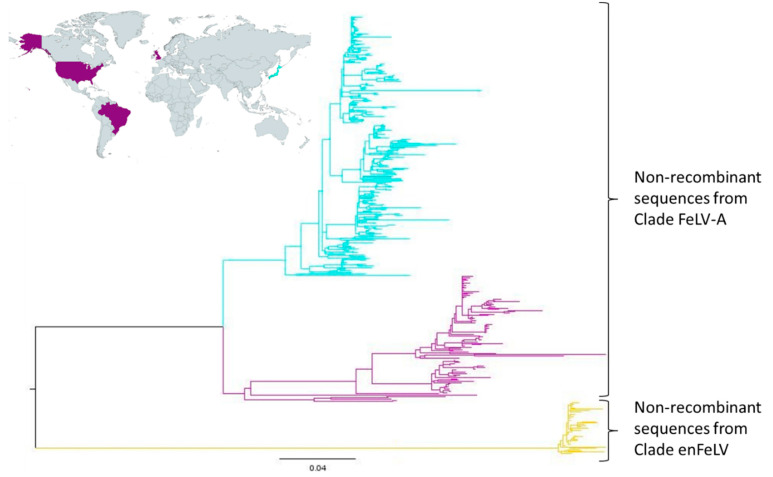
Phylogenetic reconstruction of the non-recombinant FeLV SU sequences and geographical distribution of the clade FeLV-A. Maximum likelihood tree reconstructed by incorporating the best-fitted nucleotide substitution model (GTR+F+R5) in IQtree web server. The robustness of the resulting tree was evaluated by rapid bootstrapping (UFB) with 1000 pseudoreplicate datasets. The phylogenetic tree was visualized in FigTree v1.4.4 [34] and drawn using R [35]. Three main clades are presented. The yellow clade corresponds to non-recombinant sequences from clade enFeLV and the blue and purple to the non-recombinant sequences from clade A (Figure 1).

## Data Availability

Not applicable.

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
