# Peer review of "Could Phylogenetic Analysis Be Used for Feline Leukemia Virus (FeLV) Classification?"

_viruses, 2022, doi:10.3390/v14020249_

Round 1
Reviewer 1 Report
The paper by Cano-Ortiz and others presents an interesting Phylogenetic analysis of FeLV in order to understand evolution and Molecular characterization of the Virus. The paper has strong Phylogenetic support and conclusion are adequate to the presented data. Some changes are necessary for publication.
Major concerns:
Line 2: Title do not reflect the paper. It must be modified.
According to the current title a conclusion about the low utility of Phylogenetics to accurately classify FeLV Subtypes must be included and the role of recombination analysis in the understanding of FeLV subtype evolution.
Minor Concerns.
Line 41: Cite missing
Supplementary Figure 1 must be included as figure 2 at the end of line 138. Rename other figures.
Line 160: Delete references from results. If necessary, move to Discussion.
Line 166: Correct typho. “Circl”
Line 208 - Figure 3. Map is too big and Phylogeny too small. Modify according.
Line 245. Reference is needed to support this hypothesis.
Discussion. Geographical distribution of FeLV sequences is naturally biased due to the origin of the limited geographic origin of the reported sequences. Please discuss.
Line 242: Although is possible to assume that “the most recent common ancestor of clade FeLV-A presented a gene organization related to FeLV-A and used feThTr1 as its host receptor”, Author do not show Phylodynamic analysis to support this. A short paragraph about it should be included discussing a probable MRCA.
Author Response
Reviewers' comments
Reviewer #1
The paper by Cano-Ortiz and others presents an interesting Phylogenetic analysis of FeLV in order to understand evolution and Molecular characterization of the Virus. The paper has strong Phylogenetic support and conclusion are adequate to the presented data. Some changes are necessary for publication.
Major concerns:
Line 2: Title do not reflect the paper. It must be modified.
According to the current title a conclusion about the low utility of Phylogenetics to accurately classify FeLV Subtypes must be included and the role of recombination analysis in the understanding of FeLV subtype evolution.
Response: After carefully considering changing the title, the authors agreed that the current title best reflects the content of the paper at hand. The question acted as our research question since the beginning of the study and highlights the main objective of the paper. In addition, this is the first study to propose a phylogenetic approach to be used in the FeLV classification
Minor Concerns.
Line 41: Cite missing
Response: Citation included.
Supplementary Figure 1 must be included as figure 2 at the end of line 138. Rename other figures.
Response: Authors agreed and added the figure to the main body of the manuscript as Figure 2.
Line 160: Delete references from results. If necessary, move to Discussion.
Response: From the Results, we excluded the following sentence: “VRA, VRC and VRB are collectively defined as the receptor binding domain (RBD) and are the main determinants for the host cell tropism [38–41].”
Line 166: Correct typo. “Circl”
Response: We have corrected the typo.
Line 208 - Figure 3. Map is too big and Phylogeny too small. Modify according.
Response: Authors agreed, we have improved the figure.
Line 245. Reference is needed to support this hypothesis.
Response: According to the phylogenetic analysis described in this study, FeLV-C, FeLV-E and FeLV-T sequences are part of the Clade FeLV-A. However, these sequences are not grouping together within the clade suggesting that they most likely have been originated through mutation and/or recombination in separate events from FeLV-A ancestors (Figure 3). We preferred not to include a citation after this statement since this is the first time it has been said in a proper evolutionary study.
Discussion. Geographical distribution of FeLV sequences is naturally biased due to the origin of the limited geographic origin of the reported sequences. Please discuss.
Response: Authors agreed with the comment. We added the following paragraph to the results section:
“After removing the recombinant sequences from the phylogeny (Figure 4), clade FeLV-A exhibits two well-defined supported clades. Curiously, these clades group sequences from different geographical origins: while the superior clade includes only sequences from Japan, the second clade include sequences from Brazil, United States or United Kingdom. On the other hand, the non-recombinant Clade enFeLV mostly grouped sequences from Japan and China (Table S1). Unfortunately, FeLV sequences are not collected systematically throughout the world preventing further analysis about the relationship between phylogenetic classification and geographical distribution.”
In addition, we have included the following sentence to the Discussion section:
“In addition, it seems that the evolution of sequences inside clade FeLV-A is somehow related to its geographical origin, however, the lack of data in regard to the sampling location of most of the sequences analyzed here prevent further conclusions.”
Line 242: Although is possible to assume that “the most recent common ancestor of clade FeLV-A presented a gene organization related to FeLV-A and used feThTr1 as its host receptor”, Author do not show Phylodynamic analysis to support this. A short paragraph about it should be included discussing a probable MRCA.
Response: Unfortunately, the lack of information about the host receptor prevents us to do any phylodynamic analysis. In addition, as we stated in the manuscript, it is “possible to assume” that the most recent common ancestor of the Clade FeLV-A presented a gene organization related to FeLV-A since all sequences inside the Clade FeLV-A (n=315) have the same genetic pattern (Figure 3). However, we chose not to extend the discussion of MRCA due to the lack of information.
Reviewer 2 Report
“Could phylogenetic analysis be used for feline leukemia virus (FeLV) classification?” presented by Cano-Ortiz and collaborators is a well written manuscript with very insightful data into FeLV evolution. The experiments presented in this study seem to be performed carefully and the results and figures are clear. The manuscript is therefore worthwhile to be published in the Viruses. In addition, based on the authors' previous publications, I noticed that the research group has consolidated experience regarding phylogenetic analyzes and genomic recombination analyses.
Minor revisions
Line 41: citation [5] is missing
Line 118: “New 118 complete FeLV SU sequences were also deposited in GenBank (accession MW762576 - MW762583).”. Were these sequences obtained by your research group? If so, I suggest incorporating it in the methods section.
Line 205: “USA, Brazil, United Kingdom and NA. All the Chinese sequences clustered in the enFeLV”. NA refers to not available info?
Discussion: The discussion is extremely well written, despite the depth of the data presented, clearly discussing and presenting the findings, especially for an audience less familiar with the topic of molecular phylogeny and recombination analyses. However, there is no mention of the results section “3.7. Geographical distribution of FeLV-A clade”. Please consider discussing these results, clearly stating the limitations of the analyses.
General: please consider standardizing reconstructed rather than constructed in phylogenetic trees.
GeneraI: The supplementary Figure 1 to be extremely relevant to the text, and I suggest it be presented in the main text. Please consider this option.
Author Response
Reviewers' comments
Reviewer 2
“Could phylogenetic analysis be used for feline leukemia virus (FeLV) classification?” presented by Cano-Ortiz and collaborators is a well written manuscript with very insightful data into FeLV evolution. The experiments presented in this study seem to be performed carefully and the results and figures are clear. The manuscript is therefore worthwhile to be published in the Viruses. In addition, based on the authors' previous publications, I noticed that the research group has consolidated experience regarding phylogenetic analyzes and genomic recombination analyses.
Minor revisions
Line 41: citation [5] is missing]
Response: Citation included.
Line 118: “New 118 complete FeLV SU sequences were also deposited in GenBank (accession MW762576 - MW762583).”. Were these sequences obtained by your research group? If so, I suggest incorporating it in the methods section.
Response: We sequenced 8 samples to include in this study. However, we assume that this information is of no relevance to the analysis or to the conclusion brought by this research. In order to maintain a more simplistic paper focused on the in silico analysis we decided to exclude the sentence “New complete FeLV SU sequences were also deposited in GenBank (accession MW762576 - MW762583)”.
Line 205: “USA, Brazil, United Kingdom and NA. All the Chinese sequences clustered in the enFeLV”. NA refers to not available info?
Response: To improve the manuscript, we have excluded the above-mentioned sentence and made some changes to the paragraph:
“A second tree was reconstructed using only non-recombinant FeLV sequences (Table S1). In addition to the enFeLV basal clade, this phylogeny suggests that clade FeLV-A is divided into two well-defined supported groups. Curiously, these clades group sequences from different geographical origins: while the superior clade includes only sequences from Japan, the second clade include sequences from Brazil, United States or United Kingdom. On the other hand, the non-recombinant Clade enFeLV mostly grouped sequences from Japan and China (Figure 4, Supplementary Table 1). Unfortunately, FeLV sequences are not collected systematically throughout the world preventing further conclusions about the relationship between phylogenetic classification and geographical distribution.”
Discussion: The discussion is extremely well written, despite the depth of the data presented, clearly discussing and presenting the findings, especially for an audience less familiar with the topic of molecular phylogeny and recombination analyses. However, there is no mention of the results section “3.7. Geographical distribution of FeLV-A clade”. Please consider discussing these results, clearly stating the limitations of the analyses.
Response: Unfortunately, FeLV sequences are not collected systematically throughout the world preventing further analysis about the relationship between phylogenetic classification and geographical distribution. We have improved the description of these results in The Results section:
“After removing the recombinant sequences from the phylogeny (Figure 4), clade FeLV-A exhibits two well-defined supported clades. Curiously, these clades group sequences from different geographical origins: while the superior clade includes only sequences from Japan, the second clade include sequences from Brazil, United States or United Kingdom. On the other hand, the non-recombinant Clade enFeLV mostly grouped sequences from Japan and China (Table S1). Unfortunately, FeLV sequences are not collected systematically throughout the world preventing further analysis about the relationship between phylogenetic classification and geographical distribution.”
In addition, we have included the following sentence to the Discussion section:
“In addition, it seems that the evolution of sequences inside clade FeLV-A is somehow related to its geographical origin, however, the lack of data in regard to the sampling location of most of the sequences analyzed here prevent further conclusions.”
General: please consider standardizing reconstructed rather than constructed in phylogenetic trees.
Response: The word “construct” was substituted to “reconstruct” along the text.
GeneraI: The supplementary Figure 1 to be extremely relevant to the text, and I suggest it be presented in the main text. Please consider this option.
Response: Authors agreed and added the figure to the main body of the manuscript as Figure 2.